# BIOLOGICALLY-PLAUSIBLE LEARNING ALGORITHMS CAN SCALE TO LARGE DATASETS

**Will Xiao**
Department of Molecular and Cellular Biology
Harvard University
Cambridge, MA 02138, USA
xiaow@fas.harvard.edu

**Honglin Chen**
Department of Mathematics
University of California, Los Angeles
Los Angeles, CA 90095, USA
chenhonglin@g.ucla.edu

**Qianli Liao, Tomaso Poggio**
Center for Brains, Minds and Machines
Massachusetts Institute of Technology
Cambridge, MA 02139, USA
lql@mit.edu, tp@csail.mit.edu

## ABSTRACT

The backpropagation (BP) algorithm is often thought to be biologically implausible in the brain. One of the main reasons is that BP requires symmetric weight matrices in the feedforward and feedback pathways. To address this "weight transport problem" (Grossberg, 1987), two biologically-plausible algorithms, proposed by Liao et al. (2016b) and Lillicrap et al. (2016), relax BP's weight symmetry requirements and demonstrate comparable learning capabilities to that of BP on small datasets. However, a recent study by Bartunov et al. (2018) finds that although feedback alignment (FA) and some variants of target-propagation (TP) perform well on MNIST and CIFAR, they perform significantly worse than BP on ImageNet. Here, we additionally evaluate the sign-symmetry (SS) algorithm (Liao et al., 2016b), which differs from both BP and FA in that the feedback and feedforward weights do not share magnitudes but share signs. We examined the performance of sign-symmetry and feedback alignment on ImageNet and MS COCO datasets using different network architectures (ResNet-18 and AlexNet for ImageNet; RetinaNet for MS COCO). Surprisingly, networks trained with sign-symmetry can attain classification performance approaching that of BP-trained networks. These results complement the study by Bartunov et al. (2018) and establish a new benchmark for future biologically-plausible learning algorithms on more difficult datasets and more complex architectures.

## 1 INTRODUCTION

Deep learning models today are highly successful in task performance, learning useful representations, and even matching representations in the brain (Yamins et al., 2014; Schrimpf et al., 2018). However, it remains a contentious issue whether these models reflect how the brain learns. Core to the problem is the fact that backpropagation, the learning algorithm underlying most of today's deep networks, is difficult to implement in the brain given what we know about the brain's hardware (Crick 1989; however, see Hinton 2007). One main reason why backpropagation seems implausible in the brain is that it requires sharing of feedforward and feedback weights. Since synapses are unidirectional in the brain, feedforward and feedback connections are physically distinct. Requiring them to shared their weights, even as weights are adjusted during learning, seems highly implausible.

One approach to addressing this issue is to relax the requirement for weight-symmetry in error backpropagation. Surprisingly, when the feedback weights share only the sign but not the magnitude of the feedforward weights (Liao et al., 2016b) or even when the feedback weights are random (but fixed) (Lillicrap et al., 2016), they can still guide useful learning in the network, with performance comparable to and sometimes even better than performance of backpropagation, on datasets such

as MNIST and CIFAR. Here, we refer to these two algorithms, respectively, as "sign-symmetry" and "feedback alignment." Since weight symmetry in backpropagation is required for accurately propagating the derivative of the loss function through layers, the success of asymmetric feedback algorithms indicates that learning can be supported even by inaccurate estimation of the error derivative. In feedback alignment, the authors propose that the feedforward weights learn to align with the random feedback weights, thereby allowing feedback to provide approximate yet useful learning signals (Lillicrap et al., 2016).

However, a recent paper by Bartunov et al. (2018) finds that feedback alignment and a few other biologically-plausible algorithms, including variants of target propagation, do not generalize to larger and more difficult problems such as ImageNet (Deng et al., 2009) and perform much worse than backpropagation. Nevertheless, the specific conditions Bartunov et al. tested are somewhat restrictive. They only tested locally-connected networks (i.e., weight sharing is not allowed among convolution filters at different spatial locations), a choice that is motivated by biological plausibility but in practice limits the size of the network (without weight sharing, each convolutional layer needs much more memory to store its weights), making it unclear whether poor performance was attributable solely to the algorithm, or to the algorithm on those architectures.[1] Second, Bartunov et al. did not test sign-symmetry, which may be more powerful than feedback alignment since sign-symmetric feedback weights may carry more information about the feedforward weights than the random feedback weights used in feedback alignment.

In this work, we re-examine the performance of sign-symmetry and feedback alignment on ImageNet and MS COCO datasets using standard ConvNet architectures (i.e., ResNet-18, AlexNet, and RetinaNet). We find that sign-symmetry can in fact train networks on both tasks, achieving similar performance to backpropagation on ImageNet and reasonable performance on MS COCO. In addition, we test the use of backpropagation exclusively in the last layer while otherwise using feedback alignment, hypothesizing that in the brain, the classifier layer may not be a fully-connected layer and may deliver the error signal through some other unspecified mechanism. Such partial feedback alignment can achieve better performance (relative to backpropagation) than in Bartunov et al. (2018). Taken together, these results extend previous findings and indicate that existing biologically-plausible learning algorithms remain viable options both for training artificial neural networks and for modeling how learning can occur in the brain.

## 2 METHODS

Consider a layer in a feedforward neural network. Let $x_i$ denote the input to the $i^{\text{th}}$ neuron in the layer and $y_j$ the output of the $j^{\text{th}}$ neuron. Let $W$ denote the feedforward weight matrix and $W_{ij}$ the connection between input $x_i$ and output $y_j$. Let $f$ denote the activation function. Then, Equation 1 describes the computation in the feedforward step. Now, let $B$ denote the feedback weight matrix and $B_{ij}$ the feedback connection between output $y_j$ and input $x_i$, and let $f'$ denote the derivative of the activation function $f$. Given the objective function $E$, the error gradient $\frac{\partial E}{\partial x_i}$ calculated in the feedback step is described by Equation 2.

$$y_j = f(\sigma_j), \quad \sigma_j = \sum_i W_{ij} x_i \tag{1}$$

$$\frac{\partial E}{\partial x_i} = \sum_j B_{ij} f'(\sigma_j) \frac{\partial E}{\partial y_j} \tag{2}$$

Standard backpropagation requires $B = W$. Sign-symmetry (Liao et al., 2016b) relaxes the above symmetry requirement by letting $B = \text{sign}(W)$, where $\text{sign}(\cdot)$ is the (elementwise) sign function. Feedback alignment (Lillicrap et al., 2016) uses a fixed random matrix as the feedback weight matrix $B$. Lillicrap et al. showed that through training, $W$ is adjusted such that on average, $\mathbf{e}^T W B \mathbf{e} > 0$, where $\mathbf{e}$ is the error in the network's output. This condition implies that the error correction signal $B\mathbf{e}$ lies within $90°$ of $\mathbf{e}^T W$, the error calculated by standard backpropagation.

We implement both algorithms in PyTorch for convolutional and fully-connected layers and post the code at `https://github.com/willwx/sign-symmetry`.

---

[1]Moreover, theoretical and experimental results of Poggio et al. (2017) suggest that weight-sharing is not the main reason for the good performance of ConvNets, at least when trained with backpropagation.

## 3 RESULTS

### 3.1 SIGN-SYMMETRY PERFORMS WELL ON IMAGENET

#### TRAINING DETAILS

We trained ResNet-18 (He et al., 2016) on ImageNet using 5 different training settings: 1) backprop-agation; 2) sign-symmetry for convolutional layers and backpropagation for the last, fully-connected layer; 3) sign-symmetry for all (convolutional and fully-connected) layers; 4) feedback alignment for convolutional layers and backpropagation for the fully-connected layer; and 5) feedback alignment for all (convolutional and fully-connected) layers. In sign-symmetry, at each backward step, feed-back weights were taken as the signs of the feedforward weights, scaled by the same scale $\lambda$ used to initialize that layer.[2] In feedback alignment, feedback weights were initialized once at the beginning as random variables from the same distribution used to initialize that layer. For backpropagation, standard training parameters were used (SGD with learning rate 0.1, momentum 0.9, and weight decay $10^{-4}$). For ResNet-18 with other learning algorithms, we used SGD with learning rate $0.05^3$, while momentum and weight decay remain unchanged. For AlexNet with all learning algorithms, standard training parameters were used (SGD with learning rate 0.01, momentum 0.9, and weight decay $5 \times 10^{-4}$). We used a version of AlexNet (Krizhevsky, 2014, as used in `torchvision`) which we slightly modified to add batch normalization (Ioffe & Szegedy, 2015) before every non-linearity and consequently removed dropout. For all experiments, we used a batch size of 256, a learning rate decay of 10-fold every 10 epochs, and trained for 50 epochs.

#### RESULTS

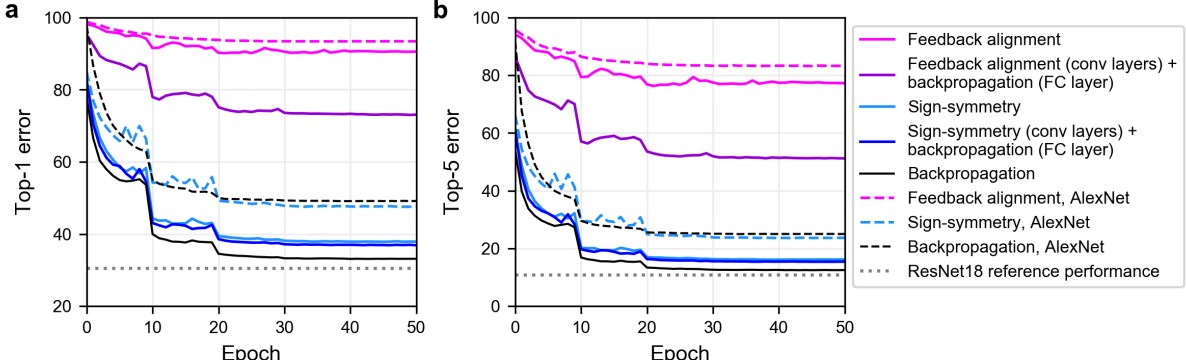

Figure 1: **a**, Top-1 and **b**, top-5 validation error on ImageNet for ResNet-18 and AlexNet trained with different learning algorithms. Dashed lines, ResNet-18 reference performance (Johnson et al., 2016). Sign-symmetry performed nearly as well as backpropagation, while feedback alignment performed better than previously reported when backpropagation was used to train the last layer.

In all cases, the network was able to learn (Figure 1, Table 1). Remarkably, sign-symmetry only slightly underperformed backpropagation in this benchmark large dataset, despite the fact that sign-symmetry does not accurately propagate either the magnitude or the sign of the error gradient. Hence, this result is not predicted by the performance of signSGD (Bernstein et al., 2018), where weight updates use the sign of the gradients, but gradients are still calculate accurately; or XNOR-Net (Rastegari et al., 2016), where both feedforward and feedback weights are binary but symmet-rical, so error backpropagation is still accurate. An intuitive explanation for this performance is that the skip-connections in ResNet help prevent the degradation of the gradient being passed through many layers of sign-symmetric feedback. However, sign-symmetry also performed similarly well

---

[2]For conv layers, $\lambda = \sqrt{2/(n_{\text{kernel width}} \cdot n_{\text{kernel height}} \cdot n_{\text{output channels}})}$; for fully-connected layers, $\lambda = 1/\sqrt{n_{\text{output}}}$. In this case, each entry in the feedback matrix has the same magnitude. We have also tested random fixed magnitudes and observe similar performance.

[3]Selected as the best from (0.1, 0.05, 0.01, 0.001, 0.0001)

Table 1: ImageNet 1-crop validation accuracy of networks trained with different algorithms, all 50 epochs. BP: backpropagation; FA: feedback alignment; SS: sign-symmetry.

| Architecture & Algorithm | Top-1 Val Error, % | Top-5 Val Error, % |
|---|---|---|
| ResNet-18, FA | 90.52 | 77.32 |
| ResNet-18, FA + last layer BP | 73.01 | 51.24 |
| ResNet-18, SS | 37.91 | 16.18 |
| ResNet-18, SS + last layer BP | 37.01 | 15.44 |
| ResNet-18, BP | 33.14 | 12.49 |
| AlexNet, FA | 93.45 | 83.29 |
| AlexNet, SS | 47.57 | 23.68 |
| AlexNet, BP | 49.15 | 25.01 |

to backpropagation in a (modified) AlexNet architecture, which did not contain skip connections. Therefore, skip-connections alone do not explain the performance of sign-symmetry.

In addition, although its performance was considerably worse, feedback alignment was still able to guide better learning in the network than reported by Bartunov et al. (2018, their Figure 3) if we use backpropagation in the last layer. This condition is not unreasonable since, in the brain, the classifier layer is likely not a soft-max classifier and may deliver error signals by a different mechanism. We also tested using backpropagation exclusively for the last layer in a network otherwise trained with sign-symmetry, but the effect on the performance was minimal. One possibility why sign-symmetry performed better than feedback alignment is that in sign-symmetry, the feedback weight always tracks the sign of the feedforward weight, which may reduce the burden on the feedforward weight to learn to align with the feedback weight.

Finally, in Liao et al. (2016b), Batch-Manhattan (BM) SGD was proposed as a way to stabilize training with asymmetric feedback algorithms. In our experience, standard SGD consistently worked better than BM for sign-symmetry, but BM may improve results for feedback alignment. We have not comprehensively characterized the effects of BM since many factors like learning rate can affect the outcome. Future experiments are needed to draw stronger conclusions.

## 3.2 MICROSOFT COCO DATASET

Besides the ImageNet classification task, we examined the performance of sign-symmetry on the MS COCO object detection task. Object detection is more complex than classification and might therefore require more complicated network architecture in order to achieve high accuracy. Thus, in this experiment we assessed the effectiveness of sign-symmetry in training networks that were more complicated and difficult to optimize.

### TRAINING DETAILS

We trained the state-of-the-art object detection network RetinaNet proposed by Lin et al. (2018) on the COCO `trainval35k` split, which consists of 80k images from `train` and 35k random images from the 40k-image `val` set. RetinaNet comprises a ResNet-FPN backbone, a classification subnet, and a bounding box regressing subnet. The network was trained with three different training settings: 1) backpropagation for all layers; 2) backpropagation for the last layer in both subnets and sign-symmetry for rest of the layers; 3) backpropagation for the last layer in both subnets and feedback alignment for rest of the layers. We used a backbone ResNet-18 pretrained on ImageNet to initialize the network. In all the experiments, the network was trained with SGD with an initial learning rate of 0.01, momentum of 0.9, and weight decay of 0.0001. We trained the network for 40k iterations with 8 images in each minibatch. The learning rate was divided by 10 at iteration 20k.

### RESULTS

The results on COCO are similar to those on ImageNet, although the performance gap between SS and BP on COCO is slightly more prominent (Figure 2). A number of factors could have potentially contributed to this result. We followed the Feature Pyramid Network (FPN) architecture design choices, optimizers, and hyperparameters reported by Lin et al. (2018); these choices are all

optimized for use with backpropagation instead of sign-symmetry. Hence, the results here represent a lowerbound on the performance of sign-symmetry for training networks on the COCO dataset.

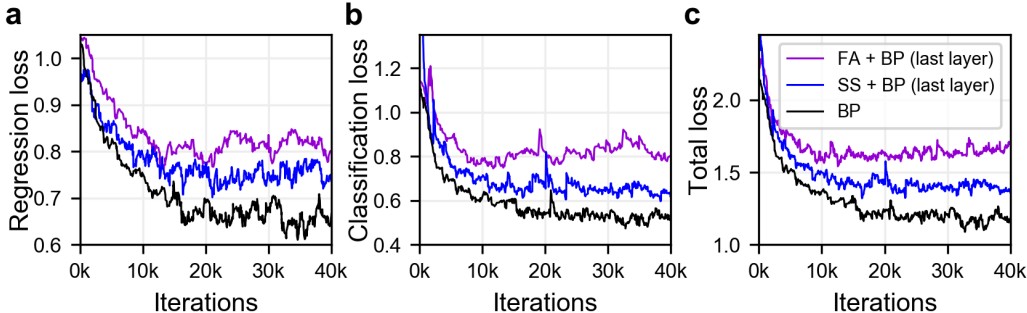

Figure 2: Training loss of RetinaNet on COCO dataset trained with 3 different settings: 1) backpropagation for all layers; 2) backpropagation for the last layer in both regression and classification subnets, and sign-symmetry for other layers; 3) backpropagation for the last layer in both subnets and feedback alignment for other layers. **a**, Object detection bounding box regression loss. **b**, Focal classification loss. **c**, Total loss.

## 4 DISCUSSION

### 4.1 COMPARING LEARNING IN SS, FA, AND BP

We ran a number of analyses to understand how sign-symmetry guides learning. Lillicrap et al. (2016) show that with feedback alignment, the alignment angles between feedforward and feedback weights gradually decrease because the feedforward weights learn to align with the feedback weights. We asked whether the same happens in sign-symmetry by computing alignment angles as in Lillicrap et al. (2016): For every pair of feedforward and feedback weight matrices, we flattened the matrices into vectors and computed the angle between the vectors. Interestingly, we found that during training, the alignment angles decreased for the last 3 layers but increased for the other layers (Figure 3a). In comparison, in the backpropagation-trained network (where $\text{sign}(W)$ was not used in any way), the analogous alignment angle between $W$ and $\text{sign}(W)$ increased for all layers. One possible explanation for the increasing trend is that as the training progresses, the feedforward weights tend to become sparse. Geometrically, this means that feedforward vectors become more aligned to the standard basis vectors and less aligned with the feedback weight vectors, which always lie on a diagonal by construction. This explanation is consistent with the similarly increasing trend of the average kurtosis of the feedforward weights (Figure 3b), which indicates that values of the weights became more dispersed during training.

Since the magnitudes of the feedforward weights were discarded when calculating the error gradients, we also looked at how sign-symmetry affected the size of the trained weights. Sign-symmetry and backpropagation resulted in weights with similar magnitudes (Figure 3c). More work is needed to elucidate how sign-symmetry guides efficient learning in the network.

### 4.2 WHY DO OUR RESULTS DIFFER FROM PREVIOUS WORK?

Our results indicate that biologically-plausible learning algorithms, specifically sign-symmetry and feedback alignment, are able to learn on ImageNet. This finding seemingly conflicts with the findings by Bartunov et al. (2018). Why do we come to such different conclusions?

First, Bartunov et al. did not test sign-symmetry, which is expected to be more powerful than feedback alignment, because it is a special case of feedback alignment that allows feedback weights to have additional information about feedforward weights. Indeed, on ImageNet, the performance of sign-symmetry approached that of backpropagation and exceeded the performance of feedback alignment by a wide margin. Another reason may be that instead of using standard ConvNets on ImageNet, Bartunov et al. only tested locally-connected networks. While the later is a more biologically plausible architecture, in practice, it is limited in size by the need to store separate weights

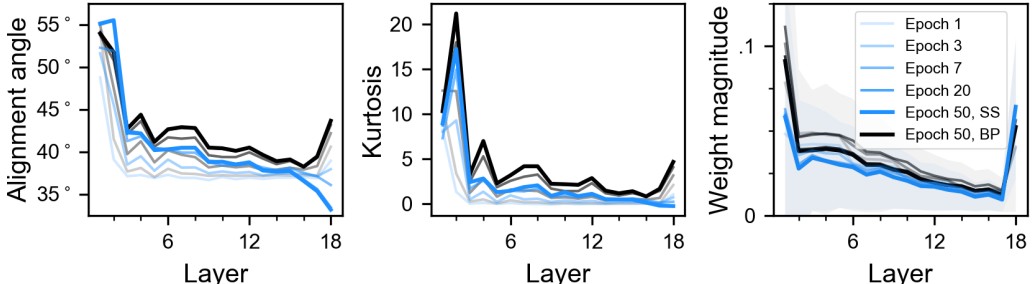

Figure 3: **a**, During training with sign-symmetry, alignment angles between feedforward weights $W$ and feedback weights $\text{sign}(W)$ decreased in the last 3 layers but increased in early layers, whereas during training with backpropagation, the analogous alignment angles increased for all layers and were overall larger. **b**, Kurtosis of the feedforward weight matrices increased during training. **c**, The magnitudes of weights trained by sign-symmetry were similar to those trained by backpropagation. Line and shading, mean $\pm$ std for epoch 50.

for each spatial location. This reduced model capacity creates a bottleneck that may affect the performance of feedback alignment (see Lillicrap et al., 2016, Supplementary Note 9). Finally, the performance of feedback alignment also benefited from the use of backpropagation in the last layer in our conditions.

### 4.3 TOWARDS A MORE BIOLOGICALLY PLAUSIBLE LEARNING ALGORITHM

A major reason why backpropagation is considered implausible in the brain is that it requires exact symmetry of physically distinct feedforward and feedback pathways. Sign-symmetry and feedback alignment address this problem by relaxing this tight coupling of weights between separate pathways. Feedback alignment requires no relation at all between feedforward and feedback weights and simply depends on learning to align the two. Hence, it can be easily realized in the brain (for example, see Lillicrap et al., 2016, Supplementary Figure 3). However, empirically, we and others have found its performance to be not ideal on relatively challenging problems.

Sign-symmetry, on the other hand, introduces a mild constraint that feedforward and feedback connections be "antiparallel": They need to have opposite directions but consistent signs. This can be achieved in the brain with two additional yet plausible conditions: First, the feedforward and feedback pathways must be specifically wired in this antiparallel way. This can be achieved by using chemical signals to guide specific targeting of axons, similar to how known mechanisms for specific wiring operate in the brain (McLaughlin & O'Leary, 2005; Huberman et al., 2008). One example scheme of how this can be achieved is shown in Figure 4. While the picture in Figure 4a is complex, most of the complexity comes from the fact that units in a ConvNet produce inconsistent outputs (i.e., both positive and negative). If the units are consistent (i.e., producing exclusively positive or negative outputs), the picture simplifies to Figure 4b. Neurons in the brain are observed to be consistent, as stated by the so-called "Dale's Law" (Dale, 1935; Strata & Harvey, 1999). Hence, this constraint would have to be incorporated at some point in any biologically plausible network, and remains an important direction for future work. We want to remark that Figure 4 is meant to indicate the relative ease of wiring sign-symmetry in the brain (compared to, e.g., wiring a network capable of weight transport), not that the brain is known to be wired this way. Nevertheless, it represents a hypothesis that is falsifiable by experimental data, potentially in the near future.[4]

Related, a second desideratum is that weights should not change sign during training. While our current setting for sign-symmetry removes weight magnitude transport, it still implicitly relies on "sign transport." However, in the brain, the sign of a connection weight depends on the type of the

---

[4]A paper from last year examined connectivity patterns within tissue sizes of approx. 500 microns and axon lengths of approx. 250 microns (Schmidt et al., 2017); recent progress (fueled by deep learning) can trace axons longer than 1 mm (Januszewski et al., 2018; Jain & Januszewski, 2018), although the imaging of large brain volumes is still limiting. In comparison, in mice, adjacent visual areas (corresponding to stages of visual processing) are 0.5–several mms apart (Marshel et al., 2011), while in primates it is tens of millimeters. Thus, testing the reality of sign-symmetric wiring is not quite possible today but potentially soon to be.

presynaptic neuron—e.g., glutamatergic (excitatory) or GABAergic (inhibitory)—a quality that is intrinsic to and stable for each neuron given existing evidence. Hence, if sign-symmetry is satisfied initially—for example, through specific wiring as just described—it will be satisfied throughout learning, and "sign transport" will not be required. Thus, evaluating the capacity of sign-fixed networks to learn is another direction for future work.

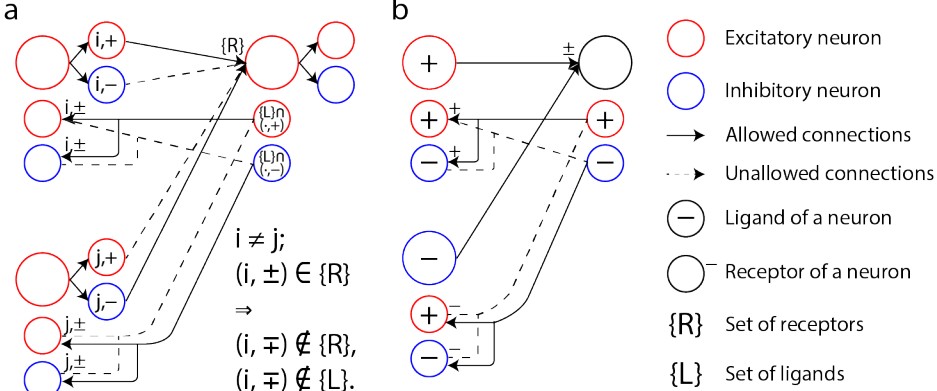

Figure 4: The specific wiring required for sign-symmetric feedback can be achieved using axonal guidance by specific receptor-ligand recognition. Assume that an axon carrying ligand $L_X$ will only synapse onto a downstream neuron carrying the corresponding receptor $R_X$. By expressing receptors and ligands in an appropriate pattern, an antiparallel wiring pattern can be established that supports sign-symmetric feedback. **a**, An example scheme. In this scheme, one inconsistent unit (i.e., a unit that produce both positive and negative outputs) in the network is implemented by three consistent biological neurons, so that each synapse is exclusively positive or negative. $n_{\text{input neurons}}$ orthogonal ligand-receptor pairs is sufficient to implement all possible connection patterns. **b**, An example scheme for implementing a sign-symmetric network with consistent units. Only 2 orthogonal ligand-receptor pairs are needed to implement all possible connectivities in this case. These schemes represent falsifiable hypotheses, although they do not exclude other possible implementations.

Another element of unclear biological reality, common to feedback alignment and sign-symmetry, is that the update of a synaptic connection (i.e., weight) between two feedforward neurons (A to B) depends on the activity in a third, feedback neuron C, whose activation represents the error of neuron B. One way it can be implemented biologically is for neuron C to connect to B with a constant and fixed weight. When C changes its value due to error feedback, it will directly induce a change of B's electric potential and thus of the postsynaptic potential of the synapse between A and B, which might lead to either Long-term Potentiation (LTP) or Long-term Depression (LTD) of synapse A-B.

Biological plausibility of ResNet has been previously discussed by Liao & Poggio (2016), claiming that ResNet corresponds to an unrolled recurrent network in the visual cortex. However, it is unclear yet how backpropagation through time can be implemented in the brain. Biological plausibility of batch normalization has been discussed in Liao et al. (2016a), where they addressed the issues with online learning (i.e., one sample at a time, instead of minibatch), recurrent architecture and consistent training and testing normalization statistics.

Other biological constraints include removing weight-sharing in convolutional layers as in Bartunov et al. (2018), incorporating temporal dynamics as in Lillicrap et al. (2016), using realistic spiking neurons, addressing the sample inefficiency general to deep learning, etc. We believe that these are important yet independent issues to the problem of weight transport and that by removing the latter, we have taken a meaningful step toward biological plausibility. Nevertheless, many steps remain in the quest for a truly plausible, effective, and empirically-verified model of learning in the brain.

## 5 CONCLUSION

Recent work shows that biologically-plausible learning algorithms do not scale to challenging problems such as ImageNet. We evaluated sign-symmetry and re-evaluated feedback alignment on their effectiveness training ResNet and AlexNet on ImageNet and RetinaNet on MS COCO. We find that

1) sign-symmetry performed nearly as well as backpropagation on ImageNet, 2) slightly modified feedback alignment performed better than previously reported, and 3) both algorithms had reasonable performance on MS COCO with minimal hyperparameter tuning. Taken together, these results indicate that biologically-plausible learning algorithms, in particular sign-symmetry, remain promising options for training artificial neural networks and modeling learning in the brain.

ACKNOWLEDGMENTS

This work was supported in part by the Center for Brains, Minds and Machines (CBMM), funded by NSF STC award CCF-1231216 and in part by C-BRIC, one of six centers in JUMP, a Semiconductor Research Corporation (SRC) program sponsored by DARPA, the National Science Foundation, Intel Corporation, and the DoD Vannevar Bush Fellowship. We gratefully acknowledge the support of NVIDIA Corporation with the donation of the DGX-1 used for this research.

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
