# OpenReview forum: "Biologically-Plausible Learning Algorithms Can Scale to Large Datasets"
_ICLR.cc/2019/Conference_

### Official Review · AnonReviewer1 · 2018-10-24
**the claims, conclusion, and general writing need to be better situated in the context of the concerns in the field**

**Rating:** 4
**Confidence:** 4

**Review:**

This work adds to a growing literature on biologically plausible (BP) learning algorithms. Building off a study by Bartunov et al. that shows the deficiencies of some BP algorithms when scaled to difficult datasets, the authors evaluate a different algorithm, sign-symmetry, and conclude that there are indeed situations in which BP algorithms can scale. This seemingly runs counter to the conclusions of Bartunov et al.; while the authors state that their results are "complementary", they also state that the findings “directly conflict” with the results of Bartunov, concluding that BP algorithms remain viable options for both learning in artificial networks and the brain.

To reach these conclusions the authors report results on a number of experiments. First, they show successful training of a ResNet-18 architecture on ImageNet using sign-symmetry, with their model performing nearly as well as one trained with backpropagation. Next, they demonstrate decent performance on MS COCO object detection using RetinaNet. Finally, they end with a discussion that seeks to explain the differences in their approach and the approach of Batunov et al, and with a potential biological implementation of sign symmetry.

Overall the clarity of the writing is sufficient. The algorithm is properly explained, and there are sufficient citations to reference prior work. The results are generally clear (though there is an incomplete experiment, I agree with the authors that it is unlikely for the preliminary results to change). I believe that there is enough detail for this work to be reproducible. The work is also sufficiently novel in that experiments using sign-symmetry on difficult datasets have not been undertaken, to my knowledge.

Unfortunately, the clarity and rigor of the *scientific argument* is insufficient for a number of reasons. These will be enumerated below.

First, the explicit writing and underlying tone of the paper reveal a misrepresentation of the scientific argument in Bartunov et al. The scientific question in Bartunov et al. is not a matter of whether BP algorithms can be useful in purely artificial settings, but rather whether they can say anything about the way in which the brain learns. In this work, on the other hand, there seems to be two scientific questions: first, to assess whether BP algorithms can be useful in artificial settings, and second, to determine whether they can say anything about how the brain learns, as in Bartunov (indeed, the author’s conclusions highlight precisely these two points). Unfortunately, the experiments and underlying experimental logic push towards addressing the first question, and use this as evidence towards a conclusion to the second question. More concretely, experiments are run on biologically problematic architectures such as ResNet-18, often with backpropagation in the final layer (though admittedly this doesn’t seem to be an important detail with sign-symmetry, for reasons explained below). This is fine under the pretense of answering the first question, but to seriously engage with the results of Bartunov et al. and assess sign-symmetry’s merit as a BP algorithm for learning in the brain, the work requires the authors the algorithms to be tested under similar conditions before claiming that there is a “direct conflict”. To this end, though the authors claim that the conditions on which Bartunov et al tested are “somewhat restrictive”, this logic can equally be flipped on its head: the conditions under which this paper tests sign-symmetry are not restrictive enough to productively move in the direction of assessing sign-symmetry’s usefulness as a description of learning in the brain, and so the conclusion that the algorithm remains a viable option for describing learning in the brain is not sufficiently supported. On the other hand, I think the conclusions regarding the first question -- whether sign-symmetry can be useful in artificial settings -- are fine given the experiments.

Second, the work does not sufficiently weigh the “degree” of implausibility of sign-symmetry compared to the other algorithms, and implicitly speaks of feedback alignment, target propagation, and sign-symmetry as equally realistic members of a class of BP algorithms. Of course, one doesn’t want to go down the road of declaring that “algorithm A is more plausible than algorithm B!”, but the nuances should at least be seriously discussed if the algorithms are to be properly compared. In backpropagation the feedback connections must be similar in sign and magnitude. Sign-symmetry eliminates the requirement that the connections be similar in magnitude. However, this factor is arguably the least important of the two (the direction of the gradient is more important than the magnitudes), and we are still left with feedback weights that somehow have to tie their sign to their feedforward counterparts, which is not an issue in target propagation or feedback alignment. The authors try to explain away this difficulty with an appeal to molecular biology, which leads into my third point.

Third, the appeal to molecular mechanisms to explain how sign-symmetry can arise is not rigorous. There is a plethora of molecular mechanisms at play in our cells; indeed, there are enough mechanisms to hand-craft *any* sort of circuit one likes. Thus, it is somewhat vacuous to conclude that a particular circuit can be “easily implemented” in the brain simply by appealing to a hand-crafted circuit. For this argument to hold one needs to appeal to biological data to demonstrate that such a circuit either a) exists already, b) most probably exists because of reasons X, Y, Z. Unfortunately there is no biological backing, rendering this argument a possibly fun thinking exercise, but not a serious scientific proposal. But perhaps most problematic, the argument leaves the problem of sign-switching in the feedforward network to “future work”. This is perhaps *the most* important problem at play here, and until it is answered, these arguments don’t have sufficient impact.

Altogether the scientific argument of this work needs tightening. The tone, the title, and the overall writing should be modified to better tackle the nuances underlying the arguments of biologically plausible learning algorithms. The claims and conclusions need to be more explicit, and the work needs to better seated in the context of both the previous literature, and the important questions at play for assessing biologically plausible learning algorithms.

---

> ### Author Response · Authors · 2018-11-08
> **Thank you very much for the review**
>
> As a quick comment, we really appreciate your very detailed feedback! We are working heavily on a hopefully equally detailed reply! :D

---

> ### Author Response · Authors · 2018-11-21
> **Thank you very much for the review**
>
> Again, thank you for the thoughtful and detailed review. We agree with most of your comments, and have edited the writing to more clearly discuss our contribution as it relates to other work in the BP arena. Please see the revised manuscript for changes to the text. As a summary, we more clearly discuss the following:
>
> 1) The significance and limitation of sign-symmetry
> As the reviewer points out, the present SS algorithm removes weight magnitude transport but still requires forward and backward weights to communicate their signs. We think it was not so clear that sign of the feedback weights is more important than magnitude before this work. While the direction of the gradient is sufficient for training as demonstrated by signSGD and related work, what we retain is not the sign of the gradient but rather the sign of the weights. As a consequence, as the error signal is propagated down the layers, the gradient may lose not only its magnitude but also its sign as compared to the backpropagated gradient. Consider the following simple example: an input is connected respectively by weights {1, -0.5} to two outputs receiving gradient {1, 1.5}. The gradient on the input computed by BP will be 0.25; that computed by SS will be -0.5. They do not share their direction! Hence, SS is not simply a coarser gradient update, but represents coarser error propagation that is nonetheless effective.
>
> Nevertheless, the requirement for sign communication is an additional assumption as compared to FA, representing a qualitative cost in "plausibility." We more explicitly discuss the “degree” of implausibility of sign-symmetry compared to other algorithms in the revised discussion section.
>
> 2) Molecular mechanisms
> Although we agree that biological molecular mechanisms are rich enough to implement a large variety of schemes, we still think there is something to be said about the simplicity or ease with which one scheme can be implemented compared to another--as judged by, e.g., the number of unique genes or interactions needed. Although we have not tried to devise a scheme for implementing BP, our intuition is that it will be much more difficult, if only because the information needed to be communicated is more (order of 10 bits for magnitude compared to 1 bit for sign). Moreover, in the truly biological case of consistent neurons, implementing sign-symmetry is rather easy (Figure 4b).
>
> Hence, the purpose of Figure 4 is to illustrate only that sign-symmetry can be achieved relatively *simply* in the brain (only 2 orthogonal ligand-receptor pairs in the case of Figure 4b), not that it is the only or even the most *likely* implementation. We would further like to remark that it is currently difficult to verify/falsify this sort of inter-areal wiring scheme in the brain given the current limits of connectomics. It is challenging to a) image large tissue sizes and b) trace axons over long distances. As a frame of reference, a paper from 1 yr ago examined tissue sizes of ~500 microns and axon lengths of ~250 microns (doi.org/10.1038/nature24005); recent work pushes the limit to ~1 mm (doi.org/10.1038/s41592-018-0049-4). In comparison, in mice a visual area spans 0.5-several mms (doi.org/10.1016/j.neuron.2011.12.004); in primates it is tens of millimeters. In this light, although Figure 4 is a thought experiment, it also represents a falsifiable hypothesis that, just like the proposed scheme for feedback alignment (Lillicrap et al. 2016, their Figure S3), can be tested with experimental data potentially in the near future.
>
> 3) Sign switching
> We agree that removing sign switching will probably greatly benefit sign-symmetry (since it removes the need for sign-transport). We have run experiments where weights do not switch sign, but find the preliminary results difficult to interpret and insufficient to report. On the other hand, we think sign switching is a core issue for any algorithm aiming at biological plausibility, but FA does not address it; nor do Bartunov et al. despite their carefully controlled architecture. That is not a criticism. How could addressing this point improve biological plausibility, unless we can also remove inconsistent neuron outputs (i.e., observe Dale's Law)? In general, there are many elements of biologically implausibility in current deep learning settings, as discussed in Section 4.3, "Towards a more biologically plausible learning algorithm." To make practical progress, we think it is still meaningful to make  stepwise advances. What we contribute is that imprecise error propagation (both in magnitude and sign) is still very useful for guiding learning.
>
> We are grateful for the reviewer for tracing out the nuances in the problem of biological plausibility, and hope we have sufficiently incorporated them into the revised manuscript and tightened our argument.

---

### Official Review · AnonReviewer2 · 2018-11-02
**Nice alternative to backprop**

**Rating:** 9
**Confidence:** 4

**Review:**

In the submitted manuscript, the authors compare the performance of sign-symmetry and feedback alignment on ImageNet and MS COCO datasets using different network architectures, with the aim of testing biologically-plausible learning algorithms alternative to the more artificial backpropagation.
The obtained results are promising and quite different to those in (Bartunov , 2018) and lead to the conclusion that biologically plausible learning algorithms in general and sign- symmetry in particular are effective alternatives for ANN training.

Although all the included ideas are not fully novel, the manuscript shows a relevant originality, paving the way for what can be a major breakthrough in deep learning theory and practice in the next few years. The paper is well written and organised, with the tackled problem well framed into the context. The suite of experiments is broad and diverse and overall convincing, even if the performances are not striking. Very interesting the biological interpretation and the proposal for the construction in the brain.
A couple of remarks: I would be interested in understanding the robustness of the sign-symmetry algorithm w.r.t. for instance dropout and (mini)batch size, and to see the behaviour of the algorithm on datasets with small sample size; second, there is probably too much stress on comparing w/ (Bartunov , 2018), while the manuscript is robust enough not to need such motivation.

Minor: refs are not homogeneous, first names citations are not consistent.

---

> ### Author Response · Authors · 2018-11-08
> **Thank you very much for the review**
>
> Thank you very much for your encouraging review.
>
> Regarding datasets with small sample size, many such experiments can be found in [1]. We did not formally repeat them but observe similar conclusions.
>
> This is a quick reply and we are working on a more detailed version!
>
> [1] Liao, Q., Leibo, J. Z., & Poggio, T. (2015). How Important is Weight Symmetry in Backpropagation?. arXiv 2015, AAAI 2016

---

> ### Author Response · Authors · 2018-11-21
> **Thank you very much for the review**
>
> Regarding writing, we have lessened the weight placed on comparison to Bartunov et al. and have made citation styles consistent. Thank you very much for suggesting these changes.
>
> Regarding suggested experiments, due to time limits we are not able to extensively test the suggested conditions. We have tested smaller batch sizes ({128, 64} vs. 256) for several training epochs, and we observe very little difference in performance for these initial epochs; in our experience initial performance is a fairly good indicator of final performance (1 epoch is still 5-20 k iters). We have also tested the effect of dropout by reintroducing them to layers fc6 and fc7 in AlexNet (in the modified AlexNet we originally tested, we removed dropout because we used BatchNorm [https://arxiv.org/abs/1502.03167]); or, adding dropout before the fc layer in ResNet-18. In both cases, dropout led to slightly slower training in the first few epochs, although we do know whether it will lead to improved converged performance. Thank you for suggesting these conditions for better characterizing the behavior of SS.

---

### Official Review · AnonReviewer3 · 2018-11-02
**An important step in our understanding of biologically plausible learning.**

**Rating:** 9
**Confidence:** 5

**Review:**

Summary: The authors are interested in whether particular biologically plausible learning algorithms scale to large problems (object recognition and detection using ImageNet and MS COCO, respectively). In particular, they examine two methods for breaking the weight symmetry required in backpropagation: feedback alignment and sign-symmetry. They extend results of Bartunov et al 2018 (which found that feedback alignment fails on particular architectures on ImageNet), demonstrating that sign-symmetry performs much better, and that preserving error signal in the final layer (but using FA or SS for the rest) also improves performance.

The paper is clear, well motivated, and significant in that it advances our understanding of how recently proposed biologically plausible methods for getting around the weight symmetry problem work on large datasets.

In particular, I appreciated: the clear introduction and explanation of the weight symmetry problem and how it arises in the context of backprop, the thorough experiments on two large scale problems, the clarity of the presented results, and the discussion about future directions of study.

Minor comments:
- s/there/therefore in the first paragraph on page 2
- The authors claim that their conclusions "largely disagree with results from Bartunov et al 2018". I would suggest a slight rewording here: the authors' results *extend* our understanding of Bartunov et al 2018. They do not disagree in the sense that this paper also finds that feedback alignment alone is insufficient to train large models on ImageNet.
- Figure 1: I was expecting to see a curve for performance of feedback alignment on AlexNet
- Figure 1: The colors are hard to follow. For example, the two shades of purple represent the two FA models, which makes sense, but then there are two separate hues (black and blue) for the sign-symmetry models. Instead, I would suggest keeping black (or gray) for backpropagation (the baseline), and then using two hues of one color (e.g. light blue and dark blue) for the two sign-symmetry models. This would make it easier to group the related models.
- Figure 2: Would be nice if these colors (for backprop/FA/SS) matched the colors in Figure 1.
- Figure 3: Why is there such a small change in the average alignment angle (2 degrees?) I found that surprising.
- Figure 3: The right two panels would be clearer on the same panel. That is, instead of showing the std. dev. separately, show it as the spread (using error bars) on the plot with the mean. This makes it easier to get a sense if the distributions overlap or not.
- Figure 3 (b/c): Could also use the same colors for BP/SS as Figs 1 and 2.
- Figure 3 (caption): I think the blue/red labels in the caption are mixed up for panel (a).

---

> ### Author Response · Authors · 2018-11-08
> **Thank you very much for the review**
>
>
> We are very excited to see your encouraging review! We really appreciate your super detailed comments.
>
> This is a quick reply and we are working on detailed replies to all comments!

---

> ### Author Response · Authors · 2018-11-21
> **Thank you very much for the review**
>
> Thank you again for the review. We really appreciate your detailed and constructive comments.
>
> - We have applied the two suggested changes to the text.
> - Re: AlexNet with feedback alignment, we are currently testing this condition and expect to include partially completed training in the final review submission and fully completed training in the final draft. As expected, its performance is slightly lower than ResNet-18 trained with FA.
> - We have applied your excellent suggestions to Figures 1, 2, and 3.
> - Regarding the small change in alignment angle, our hypothesis is that sign-symmetry does not depend on alignment of weight matrices to learn. Instead, the difference in absolute weight magnitudes between BP- and SS-trained models suggests that something else is at play. We are still analyzing the training dynamics to understand how SS guides learning.

---

### Public Comment · (anonymous) · 2018-11-05
**Assuming weight transport?**

It's nice to see more interest around the question of biologically-motivated deep learning! I've been wondering a couple of things about the central claim of the manuscript.  As I understand it, the manuscript is aimed at examining the question of whether biologically motivated algorithms can scale to large and difficult datasets.  The abstract frames this question in particular around the problem of 'weight transport' and biologically motivated algorithms that do away with this issue.  I may be missing something, but it seems to me that the approach suggested in the manuscript still makes liberal use of weight transport.  That is, the proposed approach uses backward matrices that are constructed dynamically in terms of the forward weights via:

B = sign(W^{T})

Is this true? Even though this throws away sign information, this operation still transports lots of weight information from the forward path to backward synapses.  Thus, the approach appears to assume weight transport.

It might still be an interesting datapoint to know that backward passes constructed in this way are effective.  Though I would have said that this wasn't particularly surprising, since sign information is well known to be the crucial information for learning:  for example, aggressive gradient clipping works well in many instances and as early as the 1990s Rprop (robust prop [1]) was shown to work very effectively by discarding the magnitudes of gradients (and keeping just the sign information).

Several of the other biologically-motivated algorithms that are referenced in the manuscript aim to get rid of weight transport, e.g by learning useful backward weights (Difference Target Prop).  So, is it reasonable to compare the approach in this manuscript to other algorithms that don't use weight transport?  'Sign-symmetry' seems to exist in a very different category, in that it takes weight transport for granted.  If I understand correctly, the wiring diagram in Figure 4 is meant to suggest how why it would be ok to take weight transport for granted in the brain.  But, I would have said that existing empirical evidence speaks against this outlined implementation.  At the very least, I found myself wanting citations that would strengthen the claim.

In sum, it seems like what could be said given the evidence presented in the manuscript is that: if there were an algorithm that could successfully construct backward B matrices with the correct signs (i.e. matching sign(W^{T})) without weight transport, then this hypothetical algorithm would be successful on large scale datasets.  This is interesting in its own right, but at first blush this statement seems far from the central claim of the manuscript that existing biologically-plausible algorithms already scale to large data sets?  But I may have missed something in my reading of the work, and would be happy to be corrected on details.

[1] Martin Riedmiller und Heinrich Braun: Rprop - A Fast Adaptive Learning Algorithm. Proceedings of the International Symposium on Computer and Information Science VII, 1992

---

> ### Author Response · Authors · 2018-11-08
> **Thanks for the comment!**
>
>
> Thanks a lot for such a detailed and constructive comment.
>
> Many of the concerns are similar to:
> https://openreview.net/forum?id=SygvZ209F7&noteId=HJels3U-67&noteId=HJels3U-67
>
> And we answered some of them. We are going to provide more replies soon.

---

> ### Author Response · Authors · 2018-11-22
> **Thank you for the comment**
>
> As you point out, it is true that in our current implementation of sign-symmetry, there is still the issue of "sign transport," where feedforward weights have to inform feedback weights of their signs. We were originally motivated to test sign-symmetry because biological synapses do not switch signs: The sign (excitatory or inhibitory) of a synapse is determined by the identity of neurotransmitters, which is a fixed, intrinsic property of a synapse and indeed of all synapses emanating from the same presynaptic neuron (i.e., Dale's Law; Dale, 1935). Therefore, in the brain, sign-symmetry can be easily implemented without the need to worry about sign-transport (Figure 4b). This is why, in the discussion (Section 4.3), we suggest testing sign-fixed and sign-consistent neural networks; but they are beyond the scope of this paper.
>
> Regarding the effectiveness of sign-symmetric feedback weights, we think our results are not expected from work like Rprop. The crucial difference is that previous work uses sign information of the *gradient* but still computes gradients exactly using backpropagation/the chain rule. In contrast, because we use asymmetric feedforward/feedback weights, neither the magnitude *nor the sign* of the gradient is guaranteed to be preserved. Reusing the example from our other replies, consider an input connected by respective weights {1, -0.5} to two outputs receiving gradient {1, 1.5}. The gradient on the input computed by BP will be 0.25; that computed by SS will be -0.5. They do not share their sign! Therefore, our surprising finding is that even inaccurate propagation of error like this can still support learning; we are working on understanding how. (Unlike feedback alignment, we do not observe forward and backward weights to become more aligned, but they also start more aligned by construction (Figure 3a).)
>
> Regarding comparison to other biologically-plausible algorithms, all existing proposals of biologically-plausible algorithms do not solve all the implausibilities of backpropagation at once. Instead, they each address a subset of the issues. We made the discussion more explicit in Section 4.3 comparing algorithms on what problems they solve and what problems remain.
>
> On all three points, please also see our reply to Reviewer 1 and the public comment on 11/08.
>
> Thank you for raising these thoughtful points for discussion!

---

### Public Comment · (anonymous) · 2018-11-08
**misleading title, misleading claims, main result not novel**

The sign-symmetry method does not solve the weight transport problem. It just shows that a coarser kind of transport may be sufficient for practical purposes. The biological mechanism concocted in Figure 4 to show how it *may* be implemented is completely ad hoc and without any empirical support (one may as well concoct a similar scheme for standard backprop). Also, however that scheme is supposed to work (which is not explained clearly in the text, by the way), it has to show why the feedback weights have to be *exactly* +1 or -1, which is what the sign-symmetry algorithm assumes (again biologically completely unrealistically). The scheme only appears to show how sign consistency can be achieved, not why the weights have to be exactly +1 or -1.

As another commenter pointed out, the success of the sign-symmetry method in practical applications is also not surprising, given the success of the signSGD method: https://arxiv.org/abs/1802.04434 (a paper the authors unfortunately do not discuss or cite) and especially the success of the very similar (and even more restrictive) binary weight architectures (such as the XNOR-Net: https://arxiv.org/abs/1603.05279 and a whole slew of other work that followed it), again an entire literature not even mentioned in this paper.

In conclusion, the main claim of this paper (that "biologically plausible learning algorithms can scale to large datasets") is misleading and the main result is not novel.

---

> ### Author Response · Authors · 2018-11-08
> **signSGD**
>
>
> Thanks for the feedback!
>
> signSGD seems to be very similar (if not the same as) the "Batch Manhattan" (BM) approach first used in [1], which is discussed in this paper and [1].
>
> One central question in biologically-plausible training of neural network is how much different (from SGD) can the weight updates be while maintaining good performance. How much noise can SGD tolerate if evolution wants to implement an approximated SGD in the brain.
>
> With signSGD/BM, we can see that as long as the direction of the weight update is the same as standard SGD, the performance is quite good. As your comment said, this might be only mildly surprising.
>
> With sign-symmetry feedback, however, the gradients are propagated imprecisely *every* layer, leading to drastically different update directions in many early layers of the network.  Without the results of this paper and [1], it is much unclearer whether this drastic level of divergence from SGD can still lead to good performance.
>
> Although not completely eliminating the problem of weight transport, the results of this paper constitute an important step towards that direction, showing that this non-trivial level of discrepancy from SGD can be tolerated to achieve good performance on large-scale tasks like ImageNet. It is a good news for evolution --- it has more flexibility in implementing approximated SGD in the brain.
>
> [1] Liao, Q., Leibo, J. Z., & Poggio, T. (2015). How Important is Weight Symmetry in Backpropagation?. arXiv 2015, AAAI 2016
>
> Footnote: This is just a quick reply by one author. We are working on replies to all other comments. Thank you all very much for constructive comments!

---

> ### Author Response · Authors · 2018-11-17
> **Weight-transport, unitary weight, and XNOR-Net**
>
> Thank you very much for your thoughtful comments. Here is a more detailed reply after we've run additional experiments to address your concerns.
>
> 1) Weight-transport
> We do not claim to completely solve the problem of "transport." However, we do address weight-transport by eliminating the need to synchronize magnitude (many bits of information) and only asking forward and backward weights to share signs (1 bit of information). This requires a much looser connection between the two, and indeed makes it much easier to devise an implementation for SS in the brain. Although one could perhaps concoct a scheme to achieve precise weight symmetry, it will likely be much more difficult and complex because more information need to be shared. Related, although the implementation in Figure 4 is ad hoc, its purpose is only to show that it's relatively *simple* to implement sign-symmetry in the brain (especially with consistent neurons), not that it is *likely* implemented in this way in the brain. Hence, we chose to not grasp for speculative neuroscientific evidence and overstretch our claims for Figure 4.
>
> 2) Unitary weights
> We have run additional experiments where feedback weight B = sign(W) * R, where W is the feedforward weight, R is a random weight matrix (as in Feedback Alignment), and * is elementwise multiplication. This setting achieves similar performance to SS and BP in Figure 1, consistent with our interpretation that sign-symmetry works because of sign symmetry, not because of any special property of the weight magnitudes.
>
> 3) XNOR-Net
> Thank you for bringing up binary weight networks and signSGD for discussion. We omitted discussing them because they are not motivated by biological plausibility of the learning algorithm, and hence although they are superficially similar to SS, they are fundamentally different. SignSGD has been discussed in the previous comment; it still computes exact gradients, only using gradient signs during update. In XNOR-Net, although its feedforward and feedback weights are binary, they are still exactly symmetrical, making gradient computation exact in form.
>
> In contrast, consider this simple case in sign-symmetry, where input h_0 is connected by {w_0, w_1} to output {h_10, h_11}. If {w_0, w_1} = {1, -0.5} and grad_output = {1, 1.5}, grad_input = 0.25 in BP but -0.5 in SS. Not only is the SS gradient imprecise, it is in a different direction than the BP gradient! Hence SS is fundamentally different from both XNOR-Net and signSGD.
>
> We are grateful for all three comments above, and we will add them to the discussion in the paper to make our contribution clearer.

---

> > ### Public Comment · (anonymous) · 2018-11-27
> > **the difference between XNOR-net and sign-symmetry**
> >
> > I'm very interested in your work which stands in a biological perspective.
> > But, I couldn't follow some points of what you have said:
> > "In XNOR-Net, although its feed-forward and feedback weights are binary, they are still exactly symmetrical, making gradient computation exact in form."
> > Binary weights are symmetrical, so isn't it a good thing?
> > As mentioned by other commenters, in SS, the backprop weight are B = sign(W^{T}), although W is not binary, the B for error propagation is binary, so can we consider the XNOR-Net as a more restrict version where W is also binary?
> > Besides, what do you mean by "in form"? If it means standardization, isn't it good for computation?
> > Hope for your reply.

---

> > > ### Author Response · Authors · 2018-11-27
> > > **Practical and biologically-relevant differences**
> > >
> > > There are two issues here. First is whether the performance of XNOR-Net predicts the performance of SS. Saying “gradient computation in XNOR-Net is exact in form” means that because symmetrical (binary) weights are used in the forward and backward pass in XNOR-Net, credit assignment on the weights is still accurate, just like in regular backpropagation. In contrast, in SS, the error gradient is not guaranteed to have either the right magnitude or the right sign, as shown by the example in the previous reply.
> > >
> > > Now, from a purely practical standpoint, it is good that XNOR-Net accurately calculates the gradient and impressive that XNOR-Net has binary W in addition to binary B. However, neither of them helps address the second issue of biological plausibility. As the initial comment points out, biological synapses do not necessarily have binary weight either for W or for B. However, if W and B can freely vary, weight symmetry as in XNOR-Net (or backprop in general) cannot be guaranteed—creating the “weight transport problem”—and gradient calculation will no longer be accurate as in XNOR-Net or backpropagation. The performance of SS is thus unexpected because it does not limit W and B to be binary or require them to be wholly symmetrical (only symmetrical in sign), yet can still guide learning. Hence, although XNOR-Net is “more restricted,” this restriction actually helps guarantee weight symmetry and, in turn, accurate error estimation.
> > >
> > > On the point of biological plausibility, another concern with XNOR-Net is that during its training, each binary weight in W still uses an underlying real-valued buffer to allow fine-grained updates of W (arXiv:1603.05279v4, section “Training Binary-Weights-Networks”). This weight duality during inference and during update seems rather biologically problematic.
> > >
> > > Thank you for the comment. It is quite useful to clarify the superficial similarity between XNOR-Net and SS.

---

### Public Comment · (anonymous) · 2025-01-20
**Weight transport concern**

I have the same question similar with anonymous reviewer: "Assuming weight transport?"

Weight transport is "backprop seems to require rapid information transfer back along axons from each of its synaptic outputs" as mentioned in [1]. Although author mentioned that " The sign (excitatory or inhibitory) of a synapse is determined by the identity of neurotransmitters, which is a fixed, intrinsic property of a synapse and indeed of all synapses emanating from the same presynaptic neuron (i.e., Dale's Law; Dale, 1935). Therefore, in the brain, sign-symmetry can be easily implemented without the need to worry about sign-transport (Figure 4b). ", the problem of "weight transport" is not about "if it has exact value or the sign of synapses" , it foucus on the transmition direction. i.e. information is just allowed to transmit from presynapses to postsynapses.

So I may worry about the biological plausibility of sign-symmetry (SS) algorithm.

[1] Lillicrap T P, Cownden D, Tweed D B, et al. Random synaptic feedback weights support error backpropagation for deep learning[J]. Nature communications, 2016, 7(1): 13276.

---

### Meta-Review · Area_Chair1 · 2018-12-12
**worth discussing more**

**Confidence:** 4
**Recommendation:** Accept (Poster)

**Metareview:**

This heavily disputed paper discusses a biologically motivated alternative to back-propagation learning.   In particular, methods focussing on sign-symmetry rather than weight-symmetry are investigated and, importantly, scaled to large problems.  The paper demonstrates the viability of the approach.  If nothing else, it instigates a wonderful platform for debate.

The results are convincing and the paper is well-presented.  But the biological plausibility of the methods needed for these algorithms can be disputed.  In my opinion, these are best tackled in a poster session, following the good practice at neuroscience meetings.

On an aside note, the use of the approach to ResNet should be questioned.  The skip-connections in ResNet may be all but biologically relevant.